

# The National Eutrophication Survey: lake characteristics and historical nutrient concentrations

Jemma Stachelek[1]*, Chanse Ford[2], Dustin Kincaid[3], Katelyn King[1], Heather Miller[4], Ryan Nagelkirk[5]

5  Michigan State University, East Lansing, MI  48824, USA:
[1]Department of Fisheries and Wildlife, Natural Resources Building, 480 Wilson Rd
[2]Department of Earth and Environmental Sciences, Natural Science Building, 288 Farm Ln
[3]Department of Integrative Biology, Natural Science Building, 288 Farm Ln
[4]Department of Microbiology and Molecular Genetics, Biomedical and Physical Science Building, 567 Wilson Rd
10 [5]Department of Geography, Environment, and Spatial Sciences, Geography Building, 673 Auditorium Rd

*Correspondence to*: Jemma Stachelek (stachel2@msu.edu)



**Abstract.** Historical ecological surveys serve as a baseline and provide context for contemporary research, yet many of these records are not preserved in a way that ensures their long-term usability. The National Eutrophication Survey database is currently only available as scans of the original reports (PDF files) with no embedded character information. This limits its searchability, machine readability, and the ability of current and future scientists to systematically evaluate its contents.

These data were collected by the United States Environmental Protection Agency between 1972 and 1975 as part of an effort to investigate eutrophication in freshwater lakes and reservoirs. Although several studies have manually transcribed small portions of the database in support of specific studies, there have been no systematic attempts to transcribe and preserve the database in its entirety. Here we use a combination of automated optical character recognition and manual quality assurance procedures to make these data available for analysis. The performance of the optical character recognition protocol was

found to be linked to variation in the quality (clarity) of the original documents. For each of the four archival scanned reports, our quality assurance protocol found an error rate between 5.9 and 17%. The goal of our approach was to strike a balance between efficiency and data quality by combining hand-entry of data with digital transcription technologies. The finished database contains information on the physical characteristics, hydrology, and water quality of about 800 lakes in the contiguous United States (doi:10.5063/F1KK98R5). Ultimately, this database could be combined with more recent studies to

generate metadata analyses of water quality trends and spatial variation across the continental United States.

## 1 Introduction

Effective management of inland freshwater lakes requires an understanding of the factors that affect water quality and how these factors change over time. One of these factors, termed eutrophication, occurs when excess nutrient inputs from human activities fuels increases in algal growth which can cause hypoxia and decreases in water clarity. Eutrophication of surface

waters from increased phosphorus and nitrogen loading has been found to be positively correlated with altered land-use, especially in areas of rapid urbanization and intensive agriculture (Smith et al., 1999; Smith et al., 2014). As human populations and their impacts continue to grow, eutrophication is expected to become more widespread (Bennett et al., 2001; Taranu and Gregory-Eaves, 2008). Historical datasets are needed to allow for the tracking, understanding, and managing eutrophication in lakes and reservoirs because they serve as an important baseline for modern studies.

The U.S. Environmental Protection Agency (EPA) designed and implemented the National Eutrophication Survey (NES) to investigate the extent of eutrophication in freshwater lakes and reservoirs across the contiguous United States (US). Sampling took place in over 800 lakes and reservoirs from 1972 to 1975, and included a variety of physical, chemical, and biological metrics including data on nutrients and nutrient loading, hydrologic retention time, morphometry, and plankton community diversity. Unlike current EPA National Lake Assessments that select a random sample of lakes across the US,

the NES targeted only lakes impacted directly or indirectly by municipal sewage treatment plant discharge (USEPA, 2009). Until recently, these data were only available in their entirety as four separate scanned reports representing the northeastern



and northcentral (northeastern), eastern and southeastern (southeastern), central, and western regions of the US (Figure 1). In the remainder of the present paper we refer to the former two regions as simply the northeastern and southeastern regions.

To our knowledge, there have been no attempts to transcribe the data into a usable, searchable digital database despite its use in previous studies. For example, large portions of the dataset were used to examine large scale relationships

between residence time and phytoplankton abundance (Soballe and Kimmel, 1987). Also, it was used to predict eutrophication incidence in a Bayesian framework (Lamon and Stow, 2004). Smaller portions of the data were used to explore drivers of nutrient loading (Stomp et al., 2011; Brett and Benjamin, 2007). Yet, to our knowledge, the only study that has used the NES dataset to provide a publicly available data supplement is Stomp et al., (2011), and their data supplement was limited to a small subset of the available variables relating to phytoplankton community diversity.

The present study is the first to leverage digital transcription technologies to unlock the full NES dataset. In this paper, we describe the digital transcription of the full NES dataset with the goal of making the dataset openly accessible to the research community. We introduce and publish the data in an open format that requires no proprietary software. It can be easily downloaded, used for analysis, and amended. The provided summary statistics and figures also allow users to quickly assess the utility of the data. Finally, the code and raw data files are provided to facilitate the extraction of fields not

represented in our completed dataset (mostly phytoplankton diversity data).

## 2 Methods

Data was collected from multiple locations within the water column and included in-situ measurements as well as laboratory analysis. Flow estimates and drainage area calculations were provided by the USGS and were determined from flow gages when present. More detailed information on sampling methods, units, equipment, and accuracy can be found in the EPA

survey methods publication (USEPA, 1975). Due to historical nature of the dataset, the NES sampling design differs from more modern efforts (USEPA, 2009). For example, the original NES data was collected from four separate regions of the US over the course of four years, whereas current assessments complete nation-wide sampling in a single summer. As such, NES data values represent the median of measurements taken in the spring, summer, and fall in either 1972 (northeastern), 1973 (southeastern), 1974 (central), or 1975 (western) rather than measurements taken in a single year.

We obtained the NES archival scanned reports from the EPA National Service Center for Environmental Publications (available at: https://www.epa.gov/nscep). The data for each NES region is contained in four separate files. We extracted the data from each file using automated techniques followed by manual quality assurance and checking. To begin, we enhanced (de-noised) each file using the local adaptive filtering algorithm as provided by the Imagemagick program (v6.8.9-9, available at: https://www.imagemagick.org/). Next, we processed the enhanced files using the Tesseract optical

character recognition program (OCR) (Ooms, 2017; Smith, 2007). The output of these initial extraction steps was recorded in a set of "raw data" files where each file contains the raw unprocessed text of each document page. The contents of specific fields in the raw data were extracted to a database using the automated rules provided by the nesR software package



(Stachelek, 2017a). Finally, all values in the database were manually checked for accuracy against the original scanned reports. Inaccurate OCR outputs were hand-corrected in the final database.

We provide the final dataset in an open non-proprietary format (comma-delimited, *.csv). We generated metadata descriptions from the contents of the original scanned reports. All calculations, table construction, and figure generation were performed in R and saved as reproducible R scripts (R Core Team, 2017). Table and figure generation was accomplished with the use of the reshape2, plyr, and sp packages (Wickham, 2007; Wickham, 2011; Pebesma and Bivand, 2005).

## 3 Results

The final NES dataset contains observations from 775 lakes and the distribution of these lakes was spatially variable. Although there were more lakes measured in the northeastern and southeastern United States, the number of locations was close to evenly distributed among the remaining regions (Figure 1, Table 1). Specifically, the number of lakes sampled in each region were as follows: northeastern - 200 lakes, southeastern - 245 lakes, central - 177 lakes, and western, 153 lakes.

Although the overall number of lakes was similar among regions, they differed in the proportion of lakes classified as impoundments rather than natural lakes. For example, slightly more than half of the lakes studied (462 of 775) were classified as impoundments yet the northeastern region had only 54 impoundments and the southeastern region had 168 impoundments. Conversely, the number of natural lakes sampled in the northeastern region (146 lakes) was more than double that of any other region (77, 48, and 42, for southeastern, western, and central United States, respectively).

We observed substantial spatial variation in many of the individual lake characteristics. For example, lakes in the eastern sub-regions were generally smaller and shallower than lakes in the western sub-region (Table 2). In addition, lakes in the western sub-region generally had higher alkalinity and higher water clarity (Figure 2, 3). Lakes with particularly low alkalinity were found in coastal areas, whereas lakes with particularly high alkalinity were found in Nevada, western Washington, and parts of North Dakota. Comparisons among regions was easy for some well-sampled lake chemistry parameters such as total phosphorus but more difficult for under sampled lake chemistry parameters. A particularly extreme example of this difficulty was total nitrogen measurements in the eastern region, as this parameter was only measured for a single lake (Table 1).

The ability to examine these spatial trends was made possible by our optical character recognition procedure which had 6 - 17% accuracy depending on region and archival report scan quality. In total, we carried out approximately 5,000 hand-corrections to the automated data product as part of our manual quality control review. A total of approximately 650 lakes had values for at least 80% of the total number of variables shown in Table 1. On an individual lake basis, the most common "missing" data was nutrient loading estimates for individual point and nonpoint-source components. In many cases, this data may not actually be missing but it may have not been a component of the budget for every lake. For example, not all lakes have industrial land use, so no data is expected for these cases.



## 4 Data Availability

Original scanned reports from the EPA are available from the EPA National Service Center for Environmental Publications (https://www.epa.gov/nscep). Our cleaned and useable data are available for download at doi:10.5063/F1KK98R5. The data are provided as a zip file which contains all versions of the data including the raw and quality checked versions (Stachelek et

al., 2017b). Moreover, the R package and R code used to scrape and analyze the data are provided by Stachelek (2017a) so that the methods may be reproduced and openly available for (re)use. All figures and summary statistics were generated by R scripts available in the data supplement linked above.

## 5 Discussion

We have demonstrated an approach for rescuing historical data from scanned documents. This approach involved a two-step

process of automated data scraping followed by hand-curation and quality assurance. Overall, we found that optical character recognition was an efficient method for reducing the labor associated with transcribing analogue text records (Drinkwater et al., 2014). Unfortunately, optical character recognition technology does not have absolute accuracy. In our case, transcription was hampered by poor print and scan quality of the source paper documents. We discovered through our manual validation procedure that the OCR computations produced inaccurate values in approximately 6 - 17% of the cells in the complete

dataset (n = 4836). We expect that accuracy could be improved by experimenting with varying the window size of the local adaptive thresholding algorithm relative to the document font size. Our ability to experiment with thresholding window size was limited due to the computationally expensive nature of these extractions.

      The result of our approach was data from every lake and nearly every variable in the NES survey dataset. The only primary subset of the NES data that is not included in our final product is the phytoplankton distribution data, which has

already been digitally transcribed by Stomp et al. (2011). The results of the present study could be used to explore anthropogenic and environmental drivers of lake eutrophication as well as to verify previously documented trends. One example is the 2007 National Lakes Assessment (NLA) Report, which included a reanalysis of some of the NES study lakes (USEPA, 2009). This reanalysis considered population level trends in the NES lakes but did not consider trends in individual lakes or potential environmental drivers contributing to observed trends. On a population basis, the NLA reanalysis found

that less than 30% of the NES lakes had increased chlorophyll and phosphorus concentrations. The results of the present study could be used to verify these claims as well as to compare the NES data with more recent work such as the 2012 National Lakes Assessment. Note that sampling techniques may differ from current techniques, so care should be given when making comparisons. In addition to their utility in validating historical trends, this dataset demonstrates value in that it contains data regarding various hydrographic variables, such as water residence (retention) time, which are difficult to

estimate. Such data is critical to a variety of hydrological and water quality modelling efforts (Brett and Benjamin, 2008).

      Although our goal was to digitally transcribe the full NES dataset to facilitate studies on historical nutrient loading, it is worth noting the similarities between the present study and other scientific record digitization initiatives. Such initiatives



are common in the climate and ocean sciences, but are just starting to gain momentum in the biological sciences (Allan et al., 2011; Freeman et al., 2016). To our knowledge, the present study is the first large-scale attempt at digitization of historical limnology records. We hope that by making our analysis open and reproducible we will inspire future efforts to recover important records from the pre-digital era.

**6 Author Contributions**

All authors contributed to data quality assurance and edited the article text. JS conceived the study and implemented the optical character recognition code. CF, DK, and RN performed the data analysis and made figures. KK, HM, and JS wrote major parts of the manuscript text.

**7 Competing Interests**

The authors declare that they have no conflict of interest.

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



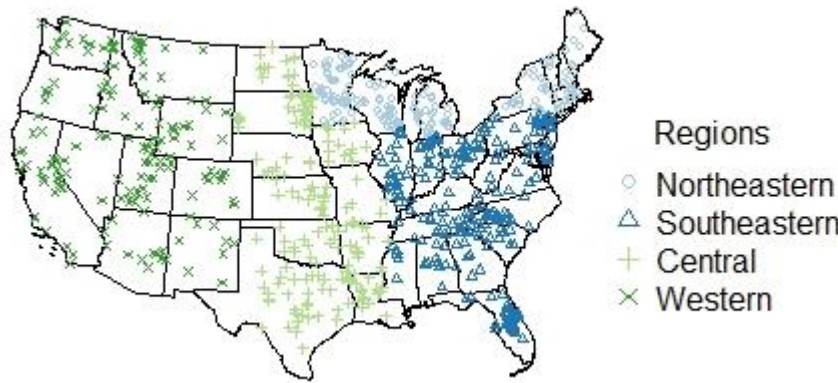

**Figure 1: Survey locations identified by sampling year (1972 northeastern-circle/light blue, 1973 southeastern-triangle/dark blue, 1974 central-plus symbol/light green, 1975 western-multiplication symbol/dark green).**

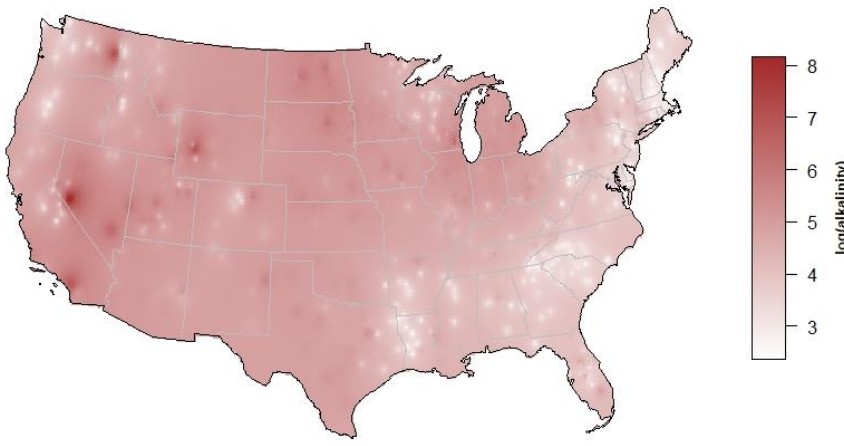

**Figure 2: Log-scaled alkalinity (mg/L) interpolated surface "heat map" generated using inverse distance weighting.**



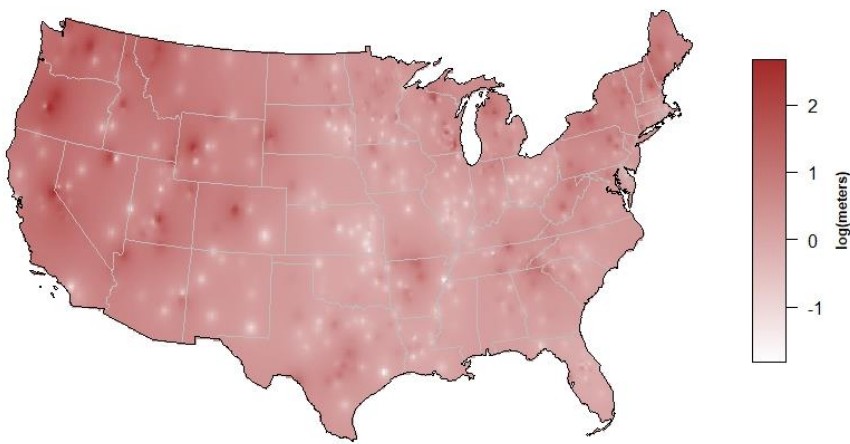

**Figure 3: Log-scaled secchi depth (m) interpolated surface "heat map" created using inverse distance weighting.**



**Table 1: Number of measurements (n) for each variable in each NES region.**

| Variable | Western | Central | Northeastern | Southeastern |
|---|---|---|---|---|
| Drainage area | 122 | 138 | 171 | 232 |
| Surface area | 152 | 177 | 200 | 245 |
| Mean depth | 149 | 174 | 174 | 242 |
| Total inflow | 124 | 138 | 170 | 232 |
| Retention time | 124 | 140 | 158 | 230 |
| Alkalinity | 153 | 177 | 200 | 245 |
| Conductivity | 153 | 176 | 200 | 245 |
| Secchi depth | 153 | 177 | 200 | 245 |
| Total P | 153 | 177 | 200 | 245 |
| Total inorg. P | 153 | 177 | 200 | 245 |
| Total inorg. N | 153 | 177 | 200 | 245 |
| Total N | 152 | 176 | 1 | 245 |
| P pt. source mun. | 52 | 83 | 139 | 189 |
| P pt. source ind. | 7 | 1 | 10 | 24 |
| P pt. source sep. | 65 | 88 | 111 | 175 |
| P nonpt. source | 122 | 133 | 167 | 231 |
| P total inputs | 122 | 133 | 167 | 231 |
| N pt. source mun. | 52 | 84 | 139 | 189 |
| N pt. source ind. | 7 | 1 | 8 | 22 |
| N pt. source sep. | 77 | 90 | 111 | 184 |
| N nonpt. source | 122 | 129 | 167 | 231 |
| N total inputs | 122 | 129 | 167 | 231 |
| P total exports | 119 | 132 | 167 | 227 |
| P retention (%) | 99 | 115 | 144 | 201 |
| P load per area | 122 | 133 | 167 | 231 |
| N total exports | 119 | 133 | 166 | 227 |
| N retention (%) | 88 | 111 | 122 | 170 |
| N load per area | 122 | 135 | 167 | 231 |



**Table 2: Mean and standard deviation (sd) for each variable in each NES region.**

| Region | Western | | | Central | | | Northeastern | | | Southeastern | | |
|---|---|---|---|---|---|---|---|---|---|---|---|---|
| Variable | Mean | | sd | Mean | | sd | Mean | | sd | Mean | | sd |
| Drainage area | 2.5e+04 | ± | 7.8e+04 | 2.1e+04 | ± | 7.5e+04 | 3.2e+03 | ± | 1.4e+04 | 5.3e+03 | ± | 1.4e+04 |
| Surface area | 44.57 | ± | 99.83 | 54.38 | ± | 1.4e+02 | 27.25 | ± | 99.01 | 42.7 | ± | 1.4e+02 |
| Mean depth | 16.71 | ± | 27.08 | 5.97 | ± | 4.49 | 7 | ± | 9.37 | 6.4 | ± | 6.07 |
| Total inflow | 52.1 | ± | 1.1e+02 | 31.82 | ± | 71.77 | 23.1 | ± | 65.26 | 82.6 | ± | 2.3e+02 |
| Retention time | 9.03 | ± | 45.25 | 8.03 | ± | 31.5 | 2.17 | ± | 5.14 | 8.78 | ± | 48.14 |
| Alkalinity | 1.7e+02 | ± | 3.7e+02 | 1.5e+02 | ± | 91.51 | 1.2e+02 | ± | 1.6e+02 | 72.18 | ± | 66.25 |
| Conductivity | 4.9e+02 | ± | 1.0e+03 | 6.4e+02 | ± | 7.6e+02 | 3.3e+02 | ± | 4.0e+02 | 2.5e+02 | ± | 2.2e+02 |
| Secchi depth | 2.86 | ± | 2.64 | 1.2 | ± | 0.91 | 1.81 | ± | 1.71 | 1.22 | ± | 0.82 |
| Total P | 0.07 | ± | 0.13 | 0.11 | ± | 0.16 | 0.16 | ± | 0.35 | 0.12 | ± | 0.27 |
| Total inorg. P | 0.04 | ± | 0.11 | 0.04 | ± | 0.07 | 0.11 | ± | 0.3 | 0.05 | ± | 0.15 |
| Total inorg. N | 0.14 | ± | 0.23 | 0.33 | ± | 0.58 | 0.47 | ± | 0.66 | 0.72 | ± | 0.91 |
| Total N | 0.62 | ± | 0.65 | 1.22 | ± | 1.11 | 0.12 | ± | NA | 1.56 | ± | 1.25 |
| P pt. source mun. | 2.5e+04 | ± | 8.7e+04 | 2.3e+04 | ± | 5.6e+04 | 3.5e+04 | ± | 1.5e+05 | 4.5e+04 | ± | 1.1e+05 |
| P pt. source ind. | 2.5e+04 | ± | 4.0e+04 | 1.3e+04 | ± | NA | 2.7e+04 | ± | 4.9e+04 | 1.7e+04 | ± | 4.5e+04 |
| P pt. source sep. | 56.62 | ± | 1.4e+02 | 60.62 | ± | 93.67 | 1.6e+02 | ± | 3.4e+02 | 98.55 | ± | 2.3e+02 |
| P nonpt. source | 1.4e+05 | ± | 4.2e+05 | 1.8e+05 | ± | 6.8e+05 | 5.6e+04 | ± | 2.1e+05 | 1.9e+05 | ± | 5.5e+05 |
| P total inputs | 1.5e+05 | ± | 4.7e+05 | 2.0e+05 | ± | 7.0e+05 | 8.7e+04 | ± | 3.4e+05 | 2.3e+05 | ± | 5.8e+05 |
| N pt. source mun. | 7.8e+04 | ± | 2.5e+05 | 7.3e+04 | ± | 1.7e+05 | 1.4e+05 | ± | 5.4e+05 | 1.4e+05 | ± | 3.8e+05 |
| N pt. source ind. | 2.3e+07 | ± | 6.1e+07 | 4.0e+03 | ± | NA | 1.6e+05 | ± | 4.2e+05 | 1.7e+05 | ± | 5.6e+05 |
| N pt. source sep. | 5.7e+06 | ± | 5.0e+07 | 2.2e+03 | ± | 3.5e+03 | 4.3e+03 | ± | 5.5e+03 | 3.3e+03 | ± | 6.7e+03 |
| N nonpt. source | 1.8e+06 | ± | 4.9e+06 | 1.8e+06 | ± | 4.4e+06 | 1.2e+06 | ± | 4.1e+06 | 3.1e+06 | ± | 8.9e+06 |
| N total inputs | 6.8e+06 | ± | 5.7e+07 | 1.8e+06 | ± | 4.3e+06 | 1.3e+06 | ± | 4.6e+06 | 3.2e+06 | ± | 9.0e+06 |
| P total exports | 6.2e+04 | ± | 1.7e+05 | 7.4e+04 | ± | 1.9e+05 | 7.3e+04 | ± | 3.1e+05 | 1.9e+05 | ± | 6.3e+05 |
| P retention (%) | 47.77 | ± | 28.5 | 56.85 | ± | 26.41 | 36.93 | ± | 25.2 | 42.7 | ± | 23.34 |
| P load per area | 5.61 | ± | 21.36 | 3.3 | ± | 9.2 | 28.46 | ± | 97.49 | 9.43 | ± | 17.06 |
| N total exports | 1.6e+06 | ± | 4.0e+06 | 1.2e+06 | ± | 2.8e+06 | 1.2e+06 | ± | 4.9e+06 | 3.0e+06 | ± | 8.3e+06 |
| N retention (%) | 39.33 | ± | 27.13 | 43.41 | ± | 23.97 | 28.41 | ± | 23.62 | 26.28 | ± | 18.85 |
| N load per area | 1.8e+02 | ± | 1.1e+03 | 42.67 | ± | 1.1e+02 | 2.8e+02 | ± | 9.1e+02 | 1.3e+02 | ± | 2.4e+02 |