# Peer review of "The National Eutrophication Survey: lake characteristics and historical nutrient concentrations"

_Earth System Science Data, 2017_

## Referee Comment (RC1) · Anonymous Referee #1 · 21 Jul 2017

This is an important data set and it is good to see that it is becoming available in electronic format. The authors appear to have used good methods to bring in a large dataset. I suspect that the manual entry error rate would be close to the lower end of their error rate.

There are multiple copies of these reports floating around in various libraries. Would it improve things to scan multiple copies of the same data and check the copies against each other?

I am curious why the authors did not try to bring in the data that Stomp et al. (2011) digitized. They could then compare the two datasets. At a minimum they should try to

combine the datasets, at least chlorophyll would be nice.

I am not functional in R so could not assess that part of the data product.

The data could use some quality checking. For example, there are points where phosphate exceeds total phosphorus. This is not possible.

It also would be good if the all data were all quality checked. If it takes about 1 second per data point, I calculate it would take 3 hours each for the team of authors working in pairs to check the whole thing. That would lead to a cleaner data set as well as making the error rate certain.

It would be nice to have retention all in one type of units (not years or days mixed)

Table 2. Could use the units

---

## Referee Comment (RC2) · Anonymous Referee #2 · 1 Nov 2017

Please use the full doi designation: https://doi.org/10.5063/f1kk98r5 ? The full designation allowed this reviewer to avoid a search through DataCite to access the KNB site.

Data very well organized and easily accessible. Very good metadata. Spot checks (Montana, Illinois) showed believable locations and values, evidently quality control has worked reasonably well.

Good product, potentially very useful as baseline for both chemical and hydrological / geomorphological purposes.

No information about sampling date in the master .csv file? E.g. a reader gets ref-

erence to the report number (.pdf 475, published 1978) and to a page number (for Bloomington Lake, Maclean County, Illinois, actually on page 79 rather than 81 as in .csv file), but no reference to sampling dates. Bloomington Lake data shows nutrient and biological samples collected on 5/11/73, 8/9/73 and 10/17/73. For MacDonald Lake (.pdf 477, page 78 rather than 80 as in .csv) Montana, nutrient and biological sampling on 6/1/75 and 7/28/75. In text we read that sampling of geographic regions occurred by year (e.g. 1973 for southeastern including Illinois and 1975 for western) but the user does not see actual dates where available, or would need to extract those dates themselves? But apparently none of the raw files captured these sampling dates from the original .pdf?

Data from Illinois resides in two separate sections of .pdf 475 (page numbers 80 to 99 and 100 to 110 contiguous) but one needs to search by storet_code or state name to find all data per each state? This scattering arises from processing sequence?

Page numbers in .csv file refer to page number of digitized .pdf, not to page numbers used within the individual reports? I did not see reference to this small discrepancy in the metadata.

Need specific clarification about the page number discrepancies and about whether the digitisation process captured the sampling dates.

---

## Author Comment (AC1) · 13 Nov 2017

**Reviewer Responses**

0.1  Reviewer 1

- *This is an important data set and it is good to see that it is becoming available in electronic format. The authors appear to have used good methods to bring in a large dataset. I suspect that the manual entry error rate would be close to the*

*lower end of their error rate.*

- *There are multiple copies of these reports floating around in various libraries. Would it improve things to scan multiple copies of the same data and check the copies against each other?*

  **This is a good suggestion. However, the effort to do this would be quite large. Also, since we manually verified the accuracy of every field, this task seems unnecessary. This may have been a more useful step at the beginning of our optical character recognition quality control step, affording us perhaps two other scan options, and the best of three could have been populated into our final repository. We would still suspect that a manual quality control step would have been necessary.**

- *I am curious why the authors did not try to bring in the data that Stomp et al. (2011) digitized. They could then compare the two datasets. At a minimum they should try to combine the datasets, at least chlorophyll would be nice.*

  **The R code provided in the article supplement could be extended to accomplish this task. It would require writing a parsing function to extract the values from the raw files produced as a result of the optical character recognition algorithm. However, this section has very complex formatting so we felt that doing so would be redundant. In addition, the Stomp et al. dataset is already archived and we felt that it was not good data management practice to duplicate their data in a second location.**

- *I am not functional in R so could not assess that part of the data product.*

- *The data could use some quality checking. For example, there are points where phosphate exceeds total phosphorus. This is not possible.*

  **We also agree that some data points reported through these four EPA documents did not make scientific sense. However, our goal was to repro-**

duce the dataset exactly as it appears in the original documents, and not to judge the scientific accuracy of these data. We expect that a proper analysis would be conducted in cooperation with an original research project. We have added text to the manuscript pointing out this issue.

- *It also would be good if the all data were all quality checked. If it takes about 1 second per data point, I calculate it would take 3 hours each for the team of authors working in pairs to check the whole thing. That would lead to a cleaner data set as well as making the error rate certain.*

  **We agree that manual quality checking is essential. We were able to calculate (and report) our error rate because we had already done this manual checking.**

- *It would be nice to have retention all in one type of units (not years or days mixed)*

  **We agree. However, if we converted the data in this way it would make it more difficult to check the data against the original documents.**

- *Table 2. Could use the units*

  **Good catch! We have updated the table.**

0.2   Reviewer 2

- *Please use the full doi designation: https://doi.org/10.5063/f1kk98r5 ? The full designation allowed this reviewer to avoid a search through DataCite to access the KNB Site.*

  **The ESSD author instructions say to report a DOI in the abbreviation form (10.5194/xyz). We suspect that this could be hyperlinked to the full designation on typesetting. For example, it appears that the partial doi code**

**provided in the abstract of the pdf manuscript has been hyperlinked to the full doi in the html abstract.**

- *Data very well organized and easily accessible. Very good metadata. Spot checks (Montana, Illinois) showed believable locations and values, evidently quality control has worked reasonably well. Good product, potentially very useful as baseline for both chemical and hydrological / geomorphological purposes.*

- *No information about sampling date in the master .csv file? E.g. a reader gets reference to the report number (.pdf 475, published 1978) and to a page number (for Bloomington Lake, Maclean County, Illinois, actually on page 79 rather than 81 as in .csv file), but no reference to sampling dates. Bloomington Lake data shows nutrient and biological samples collected on 5/11/73, 8/9/73 and 10/17/73. For MacDonald Lake (.pdf 477, page 78 rather than 80 as in .csv) Montana, nutrient and biological sampling on 6/1/75 and 7/28/75. In text we read that sampling of geographic regions occurred by year (e.g. 1973 for southeastern including Illinois and 1975 for western) but the user does not see actual dates where available, or would need to extract those dates themselves? But apparently none of the raw files captured these sampling dates from the original .pdf?*

**The sampling dates you see are exclusive to the "Biological Characteristics" section, which we did not transcribe, because this has already been done in Stomp et al. (2011). In contrast, the data we report are annual means computed from monthly samples. The NES reports provide no further details about specific sampling dates. We have added clarifying text to the manuscript and metadata on this point.**

- *Data from Illinois resides in two separate sections of .pdf 475 (page numbers 80 to 99 and 100 to 110 contiguous) but one needs to search by storet_code or state name to find all data per each state? This scattering arises from processing sequence?*

**Yes**

- *Page numbers in .csv file refer to page number of digitized .pdf, not to page numbers used within the individual reports? I did not see reference to this small discrepancy in the metadata.*

  **We have appended additional text to the "pagenum" metadata field. It now reads: "page number of the pdf (not the report page number)".**

- *Need specific clarification about the page number discrepancies and about whether the digitisation process captured the sampling date*

  **See above.**

**Supplement:**

**The National Eutrophication Survey: lake characteristics and historical nutrient concentrations**

Joseph Stachelek[1], Chanse Ford[2], Dustin Kincaid[3], Katelyn King[1], Heather Miller[4], and Ryan Nagelkirk[5]

[1]Department of Fisheries and Wildlife, Michigan State University, East Lansing, MI, USA
[2]Department of Earth and Environmental Sciences, Michigan State University, East Lansing, MI, USA
[3]Department of Integrative Biology, Michigan State University, East Lansing, MI, USA
[4]Department of Microbiology and Molecular Genetics, Michigan State University, East Lansing, MI, USA
[5]Department of Geography, Environment, and Spatial Sciences, Michigan State University, East Lansing, MI, USA

*Correspondence to:* Joseph Stachelek (stachel2@msu.edu)

**Abstract.** Historical ecological surveys serve as a baseline and provide context for contemporary research, yet many of these records are not preserved in a way that ensures their long-term usability. The National Eutrophication Survey (NES) database is currently only available as scans of the original reports (PDF files) with no embedded character information. This limits its searchability, machine readability, and the ability of current and future scientists to systematically evaluate its contents.
5   The NES data were collected by the United States Environmental Protection Agency between 1972 and 1975 as part of an effort to investigate eutrophication in freshwater lakes and reservoirs. Although several studies have manually transcribed small portions of the database in support of specific studies, there have been no systematic attempts to transcribe and preserve the database in its entirety. Here we use a combination of automated optical character recognition and manual quality assurance procedures to make these data available for analysis. The performance of the optical character recognition protocol was found
10  to be linked to variation in the quality (clarity) of the original documents. For each of the four archival scanned reports, our quality assurance protocol found an error rate between 5.9 and 17%. The goal of our approach was to strike a balance between efficiency and data quality by combining hand-entry of data with digital transcription technologies. The finished database contains information on the physical characteristics, hydrology, and water quality of about 800 lakes in the contiguous United States (doi:10.5063/F10G3H3Z). Ultimately, this database could be combined with more recent studies to generate  meta-analyses
15  of water quality trends and spatial variation across the continental United States.

**1   Introduction**

Effective management of inland freshwater lakes requires an understanding of the factors that affect water quality and how these factors change over time. One of these factors, termed eutrophication, occurs when excess nutrient inputs from human activities fuels increases in algal growth which can cause hypoxia and decreases in water clarity. Eutrophication of surface
20  waters from increased phosphorus and nitrogen loading has been observed in connection with altered land-use especially in areas of rapid urbanization and intensive agriculture (Smith et al., 1999, 2014). As human populations and their impacts continue to grow, eutrophication is expected to become more widespread (Bennett et al., 2001; Taranu and Gregory-Eaves,

[Figure]

**Figure 1.** Survey locations colored by sampling year (1972 northeastern-red, 1973 southeastern-green, 1974 central-blue, 1975 western-grey).

2008). Historical datasets are needed in order to track, understand, and manage eutrophication in lakes and reservoirs because they serve as an important baseline for modern studies.

The U.S. Environmental Protection Agency (EPA) designed and implemented the National Eutrophication Survey (NES) in order to investigate the extent of eutrophication in freshwater lakes and reservoirs across the contiguous United States
5 (US). Sampling took place in over 800 lakes and reservoirs from 1972 to 1975, and included a variety of physical, chemical, and biological metrics including data on nutrients and nutrient loading, hydrologic retention time, morphometry, and plankton community diversity. Each lake was sampled on a monthly basis for a period one year. Except for the phytoplankton distribution subset, which we did not transcribe (see Stomp et al., 2011), the NES data is provided as annual averages. Unlike current EPA National Lake Assessments that select a random sample of lakes across the US, the NES targeted only lakes impacted directly or
10 indirectly by municipal sewage treatment plant discharge (USEPA, 2009, 1975). Until recently, these data were only available in their entirety as four separate scanned reports representing the northeastern and northcentral (northeastern), eastern and southeastern (southeastern), central, and western regions of the US (Figure 1). In the remainder of the present paper we refer to the former two regions as simply the northeastern and southeastern regions.

To our knowledge, there have been no attempts to transcribe the data into a usable, searchable digital database despite
15 its use in previous studies. For example, large portions of the dataset were used to examine large scale relationships between residence time and phytoplankton abundance (Soballe and Kimmel, 1987). Also, it was used to predict eutrophication incidence in a Bayesian framework (Lamon and Stow 2004). Smaller portions of the data were used to explore drivers of nutrient loading (Stomp et al., 2011; Brett and Benjamin, 2007). Yet, to our knowledge, the only study to use the NES dataset and provide a publicly available data supplement is Stomp et al. (2011),  but their data supplement was limited to a small subset of the
20 available variables relating to phytoplankton community diversity.

The present study is the first to leverage digital transcription technologies to unlock the full NES dataset. In this paper, we describe the digital transcription of the full NES dataset with the goal of making the dataset openly accessible to the research community. Specifically, our objective was to exactly reproduce the contents of the original dataset rather than to evaluate its scientific integrity. We introduce and publish the data in an open format that requires no proprietary software. It can be easily

5 downloaded, used for analysis, and amended. The provided summary statistics and figures also allow users to quickly assess the utility of the data. Finally, the code and raw data files are provided to facilitate the extraction of fields not represented in our completed dataset (mostly phytoplankton diversity data).

**2 Methods**

Data was collected from multiple locations within the water column and included in-situ measurements as well as laboratory

10 analysisanalyses. Flow estimates and drainage area calculations were provided by the USGS and were determined from flow gages when present. More detailed information on sampling methods, units, equipment, and accuracy can be found in the EPA survey methods publication (USEPA, 1975). Due to historical nature of the dataset, the NES sampling design differs from more modern efforts (USEPA, 2009). For example, the original NES data was collected from four separate regions of the US over the course of four years, whereas current assessments complete nation-wide sampling in a single summer. As such, NES data

15 values represent the median mean of measurements taken in the spring, summer, and fall in either 1972 (northeastern), 1973 (southeastern), 1974 (central), or 1975 (western) rather than summer measurements taken in a single year.

We obtained the NES archival scanned reports from the EPA National Service Center for Environmental Publications (available at: https://www.epa.gov/nscep). The data for each NES region is contained in four separate files. We extracted the data from each file using automated techniques followed by manual quality assurance and checking of each value. To begin, we

20 enhanced (de-noised) each file using the local adaptive filtering algorithm as provided by the Imagemagick program (v6.8.9-9, available at: https://www.imagemagick.org/). Next, we processed the enhanced files using the Tesseract optical character recognition program (OCR) (Ooms, 2017; Smith, 2007). The output of these initial extraction steps were recorded in a set of "raw data" files where each file contains the raw unprocessed text of each document page. The contents of specific fields in the raw data were extracted to a database using the automated rules provided by the nesR software package (Stachelek, 2017).

25 Finally, all values in the database were manually checked for accuracy against the original scanned reports. Inaccurate OCR outputs were hand-corrected in the final database. Because our goal was to reproduce the data from the original reports and not to verify the technical correctness of the original data, we only changed values if they did not match the orginal data reports. For example, we did not change data from the five NES lakes that had phosphate $(PO^4)$ values exceeding their corresponding total phosphorus $(TP)$ values despite the fact that this is not physically possible ($PO^4$ is a component of $TP$).

30 We provide the final dataset in an open non-proprietary format (comma-delimited, *.csv). We In addition, we generated metadata descriptions from the contents of the original scanned reports. All calculations, table construction, and figure generation were performed in R and saved as reproducible R scripts (R Core Team, 2017). Table and figure generation was accomplished with the use of the reshape2, plyr, and sp packages (Wickham, 2016; Pebesma and Bivand, 2017).

[Figure]

**Figure 2.** Map of log-scaled alkalinity (mg/L) interpolated using inverse distance weighting.

**3 Results**

The final NES dataset contains observations from 775 lakes and the distribution of these lakes was spatially variable. Although there were more lakes measured in the northeastern and southeastern United States, the number of locations was close to evenly distributed among the remaining regions (Figure 1, Table 1). Specifically, the number of lakes sampled in each region were as follows: northeastern - 200 lakes, southeastern - 245 lakes, central - 177 lakes, and western, 153 lakes.

 In addition to differences in the total number of lakes  measured in each region, there were also differences in the proportion of lakes classified as impoundments rather than as natural lakes. For example, slightly more than half of all the lakes studied (462 of 775) were classified as impoundments yet the northeastern region had only 54 impoundments  while the southeastern region had 168 impoundments. Conversely, the number of natural lakes sampled in the northeastern region (146 lakes) was more than double that of any other region (77, 48, and 42, for southeastern, western, and central United States, respectively).

We observed substantial spatial variation in many of the individual lake characteristics. For example, lakes in the eastern sub-regions were generally smaller and shallower than lakes in the western sub-region (Table 2). In addition, lakes in the western sub-region generally had higher alkalinity and higher water clarity (Figure 2, 3). Lakes with particularly low alkalinity were found in coastal areas, whereas lakes with particularly high alkalinity were found in Nevada, western Washington, and parts of North Dakota. Comparisons among regions was easy for some well-sampled lake chemistry parameters such as total phosphorus but more difficult for undersampled lake chemistry parameters. A particularly extreme example of this difficulty was total nitrogen measurements in the eastern region, as this parameter was only measured for a single lake (Table 1).

The ability to examine these spatial trends was made possible by our optical character recognition procedure which had 6 - 17% accuracy depending on region and archival report scan quality. In total, we carried out approximately 5,000 hand-

**Table 1.** Number of measurements (n) for each variable in each NES region.

| Variable | Western | Central | Northeastern | Southeastern |
|---|---|---|---|---|
| Drainage area | 122 | 138 | 171 | 232 |
| Surface area | 152 | 177 | 200 | 245 |
| Mean depth | 149 | 174 | 174 | 242 |
| Total inflow | 124 | 138 | 170 | 232 |
| Retention time | 124 | 140 | 158 | 230 |
| Alkalinity | 153 | 177 | 200 | 245 |
| Conductivity | 153 | 176 | 200 | 245 |
| Secchi depth | 153 | 177 | 200 | 245 |
| Total P | 153 | 177 | 200 | 245 |
| Total inorg. P | 153 | 177 | 200 | 245 |
| Total inorg. N | 153 | 177 | 200 | 245 |
| Total N | 152 | 176 | 1 | 245 |
| P pt. source mun. | 52 | 83 | 139 | 189 |
| P pt. source ind. | 7 | 1 | 10 | 24 |
| P pt. source sep. | 65 | 88 | 111 | 175 |
| P nonpt. source | 122 | 133 | 167 | 231 |
| P total inputs | 122 | 133 | 167 | 231 |
| N pt. source mun. | 52 | 84 | 139 | 189 |
| N pt. source ind. | 7 | 1 | 8 | 22 |
| N pt. source sep. | 77 | 90 | 111 | 184 |
| N nonpt. source | 122 | 129 | 167 | 231 |
| N total inputs | 122 | 129 | 167 | 231 |
| P total exports | 119 | 132 | 167 | 227 |
| P retention | 99 | 115 | 144 | 201 |
| P load per area | 122 | 133 | 167 | 231 |
| N total exports | 119 | 133 | 166 | 227 |
| N retention | 88 | 111 | 122 | 170 |
| N load per area | 122 | 135 | 167 | 231 |

**Table 2.** Mean and standard deviation (sd) for each variable in each NES region.

| Region | Western | | Central | | Northeastern | | Southeastern | |
|---|---|---|---|---|---|---|---|---|
| Variable | Mean | sd | Mean | sd | Mean | sd | Mean | sd |
| Drainage area ($km^2$) | 2.5e+04 | ± 7.8e+04 | 2.1e+04 | ± 7.5e+04 | 3.2e+03 | ± 1.4e+04 | 5.3e+03 | ± 1.4e+04 |
| Surface area ($km^2$) | 44.57 | ± 99.83 | 54.38 | ± 1.4e+02 | 27.25 | ± 99.01 | 42.7 | ± 1.4e+02 |
| Mean depth ($m$) | 16.71 | ± 27.08 | 5.97 | ± 4.49 | 7 | ± 9.37 | 6.4 | ± 6.07 |
| Total inflow ($m^3 \cdot s^{-1}$) | 52.1 | ± 1.1e+02 | 31.82 | ± 71.77 | 23.1 | ± 65.26 | 82.6 | ± 2.3e+02 |
| Retention time ($years$) | 7.27 | ± 43.32 | 2.78 | ± 6.98 | 2.01 | ± 4.77 | 0.59 | ± 1.12 |
| Alkalinity ($mg \cdot l^{-1}$) | 1.7e+02 | ± 3.7e+02 | 1.5e+02 | ± 91.51 | 1.2e+02 | ± 1.6e+02 | 72.18 | ± 66.25 |
| Conductivity ($uohm$) | 4.9e+02 | ± 1.0e+03 | 6.4e+02 | ± 7.6e+02 | 3.3e+02 | ± 4.0e+02 | 2.5e+02 | ± 2.2e+02 |
| Secchi depth ($m$) | 2.86 | ± 2.64 | 1.2 | ± 0.91 | 1.81 | ± 1.71 | 1.22 | ± 0.82 |
| Total P ($mg \cdot l^{-1}$) | 0.07 | ± 0.13 | 0.11 | ± 0.16 | 0.16 | ± 0.35 | 0.12 | ± 0.27 |
| Total inorg. P ($mg \cdot l^{-1}$) | 0.04 | ± 0.11 | 0.04 | ± 0.07 | 0.11 | ± 0.3 | 0.05 | ± 0.15 |
| Total inorg. N ($mg \cdot l^{-1}$) | 0.14 | ± 0.23 | 0.33 | ± 0.58 | 0.47 | ± 0.66 | 0.72 | ± 0.91 |
| Total N ($mg \cdot l^{-1}$) | 0.62 | ± 0.65 | 1.22 | ± 1.11 | 0.12 | ± NA | 1.56 | ± 1.25 |
| P pt. source mun. ($kg \cdot yr^{-1}$) | 2.5e+04 | ± 8.7e+04 | 2.3e+04 | ± 5.6e+04 | 3.5e+04 | ± 1.5e+05 | 4.5e+04 | ± 1.1e+05 |
| P pt. source ind. ($kg \cdot yr^{-1}$) | 2.5e+04 | ± 4.0e+04 | 1.3e+04 | ± NA | 2.7e+04 | ± 4.9e+04 | 1.7e+04 | ± 4.5e+04 |
| P pt. source sep. ($kg \cdot yr^{-1}$) | 56.62 | ± 1.4e+02 | 60.62 | ± 93.67 | 1.6e+02 | ± 3.4e+02 | 98.55 | ± 2.3e+02 |
| P nonpt. source ($kg \cdot yr^{-1}$) | 1.4e+05 | ± 4.2e+05 | 1.8e+05 | ± 6.8e+05 | 5.6e+04 | ± 2.1e+05 | 1.9e+05 | ± 5.5e+05 |
| P total inputs ($kg \cdot yr^{-1}$) | 1.5e+05 | ± 4.7e+05 | 2.0e+05 | ± 7.0e+05 | 8.7e+04 | ± 3.4e+05 | 2.3e+05 | ± 5.8e+05 |
| N pt. source mun. ($kg \cdot yr^{-1}$) | 7.8e+04 | ± 2.5e+05 | 7.3e+04 | ± 1.7e+05 | 1.4e+05 | ± 5.4e+05 | 1.4e+05 | ± 3.8e+05 |
| N pt. source ind. ($kg \cdot yr^{-1}$) | 2.3e+07 | ± 6.1e+07 | 4.0e+03 | ± NA | 1.6e+05 | ± 4.2e+05 | 1.7e+05 | ± 5.6e+05 |
| N pt. source sep. ($kg \cdot yr^{-1}$) | 5.7e+06 | ± 5.0e+07 | 2.2e+03 | ± 3.5e+03 | 4.3e+03 | ± 5.5e+03 | 3.3e+03 | ± 6.7e+03 |
| N nonpt. source ($kg \cdot yr^{-1}$) | 1.8e+06 | ± 4.9e+06 | 1.8e+06 | ± 4.4e+06 | 1.2e+06 | ± 4.1e+06 | 3.1e+06 | ± 8.9e+06 |
| N total inputs ($kg \cdot yr^{-1}$) | 6.8e+06 | ± 5.7e+07 | 1.8e+06 | ± 4.3e+06 | 1.3e+06 | ± 4.6e+06 | 3.2e+06 | ± 9.0e+06 |
| P total exports ($kg \cdot yr^{-1}$) | 6.2e+04 | ± 1.7e+05 | 7.4e+04 | ± 1.9e+05 | 7.3e+04 | ± 3.1e+05 | 1.9e+05 | ± 6.3e+05 |
| P retention (%) | 47.77 | ± 28.5 | 57.55 | ± 26.01 | 36.93 | ± 25.2 | 42.7 | ± 23.34 |
| P load per area($g/m^2/day$) | 5.61 | ± 21.36 | 3.3 | ± 9.2 | 28.46 | ± 97.49 | 9.43 | ± 17.06 |
| N total exports ($kg \cdot yr^{-1}$) | 1.6e+06 | ± 4.0e+06 | 1.2e+06 | ± 2.8e+06 | 1.2e+06 | ± 4.9e+06 | 3.0e+06 | ± 8.3e+06 |
| N retention (%) | 39.33 | ± 27.13 | 43.41 | ± 23.97 | 28.41 | ± 23.62 | 26.28 | ± 18.85 |
| N load per area ($g/m^2/day$) | 1.8e+02 | ± 1.1e+03 | 42.67 | ± 1.1e+02 | 2.8e+02 | ± 9.1e+02 | 1.3e+02 | ± 2.4e+02 |

corrections to the automated data product as part of our manual quality control review. A total of approximately 650 lakes had values for at least 80% of the total number of variables shown in Table 1. On an individual lake basis, the most common "missing" data was nutrient loading estimates for individual point and nonpoint-source components. In many cases, this data

[Figure]

**Figure 3.** Map of secchi depth (m) interpolated using inverse distance weighting.

may not actually be missing but it may have not been a component of the budget for that particular lake. For example, not all lakes have industrial land use so no data is expected in these cases.

**4 Discussion**

We have demonstrated an approach for rescuing historical data from scanned documents. In particular, our approach involved
5  a two-step process of automated data scraping followed by hand-curation and quality assurance. Overall, we found that optical character recognition was an efficient method for reducing the labor associated with transcribing analog text records (Drinkwater et al., 2014). Unfortunately, optical character recognition technology does not have absolute accuracy. In our case, transcription was hampered by poor print and scan quality of the source paper documents. We discovered through our manual validation procedure that the OCR computations produced inaccurate values in approximately 6 - 17% of the cells in the com-
10  plete dataset (n = 4836). We expect that accuracy could be improved by experimenting with varying the window size of the local adaptive thresholding algorithm relative to the document font size. Our ability to experiment with thresholding window size was limited due to the computationally expensive nature of these extractions.

[revised manuscript text omitted]

R Core Team: R: A Language and Environment for Statistical Computing, R Foundation for Statistical Computing, Vienna, Austria, https://www.R-project.org/, 2017.

Smith, R.: An overview of the Tesseract OCR engine, in: Document Analysis and Recognition, 2007. ICDAR 2007. Ninth International Conference on, vol. 2, pp. 629–633, IEEE, 2007.

Smith, V. H., Tilman, G. D., and Nekola, J. C.: Eutrophication: impacts of excess nutrient inputs on freshwater, marine, and terrestrial ecosystems, Environmental pollution, 100, 179–196, 1999.

Smith, V. H., Dodds, W. K., Havens, K. E., Engstrom, D. R., Paerl, H. W., Moss, B., and Likens, G. E.: Comment: Cultural eutrophication of natural lakes in the United States is real and widespread, Limnology and Oceanography, 59, 2217–2225, 2014.

Soballe, D. and Kimmel, B.: A large-scale comparison of factors influencing phytoplankton abundance in rivers, lakes, and impoundments, Ecology, 68, 1943–1954, 1987.

Stachelek, J.: nesR: Scrape Data from National Eutrophication Survey archival PDFs, R package version 0.2, 2017.

Stachelek, J., Ford, C., Kincaid, D., King, K., Miller, H., and Nagelkirk, R.: The National Eutrophication Survey: lake characteristics and historical nutrient concentrations. Knowledge Network for Biocomplexity, https://doi.org/10.5063/F19S1P6H, 2017.

Stomp, M., Huisman, J., Mittelbach, G. G., Litchman, E., and Klausmeier, C. A.: Large-scale biodiversity patterns in freshwater phytoplankton, Ecology, 92, 2096–2107, 2011.

Taranu, Z. E. and Gregory-Eaves, I.: Quantifying relationships among phosphorus, agriculture, and lake depth at an inter-regional scale, Ecosystems, 11, 715–725, 2008.

USEPA: National Eutrophication Survey Methods 1973 - 1976 (Working Paper No. 175), Tech. rep., United State Environmental Protection Agency, Office of Research and Development, 1975.

USEPA: National Lakes Assessment: A Collaborative Survey of the Nation's Lakes, Tech. rep., United State Environmental Protection Agency, Office of Research and Development, 2009.

Wickham, H.: plyr: Tools for Splitting, Applying and Combining Data, https://CRAN.R-project.org/package=plyr, R package version 1.8.4, 2016.

**Reviewer Responses**

**0.1 Reviewer 1**

- *This is an important data set and it is good to see that it is becoming available in electronic format. The authors appear to have used good methods to bring in a large dataset. I suspect that the manual entry error rate would be close to the lower end of their error rate.*

- *There are multiple copies of these reports floating around in various libraries. Would it improve things to scan multiple copies of the same data and check the copies against each other?*

  **This is a good suggestion. However, the effort to do this would be quite large. Also, since we manually verified the accuracy of every field, this task seems unnecessary. This may have been a more useful step at the beginning of our optical character recognition quality control step, affording us perhaps two other scan options, and the best of three could have been populated into our final repository. We would still suspect that a manual quality control step would have been necessary.**

- *I am curious why the authors did not try to bring in the data that Stomp et al. (2011) digitized. They could then compare the two datasets. At a minimum they should try to combine the datasets, at least chlorophyll would be nice.*

  **The R code provided in the article supplement could be extended to accomplish this task. It would require writing a parsing function to extract the values from the raw files produced as a result of the optical character recognition algorithm. However, this section has very complex formatting so we felt that doing so would be redundant. In addition, the Stomp et al. dataset is already archived and we felt that it was not good data management practice to duplicate their data in a second location.**

- *I am not functional in R so could not assess that part of the data product.*

- *The data could use some quality checking. For example, there are points where phosphate exceeds total phosphorus. This is not possible.*

  **We also agree that some data points reported through these four EPA documents did not make scientific sense. However, our goal was to reproduce the dataset exactly as it appears in the original documents, and not to judge the scientific accuracy of these data. We expect that a proper analysis would be conducted in cooperation with an original research project. We have added text to the manuscript pointing out this issue.**

- *It also would be good if the all data were all quality checked. If it takes about 1 second per data point, I calculate it would take 3 hours each for the team of authors working in pairs to check the whole thing. That would lead to a cleaner data set as well as making the error rate certain.*

  **We agree that manual quality checking is essential. We were able to calculate (and report) our error rate because we had already done this manual checking.**

- *It would be nice to have retention all in one type of units (not years or days mixed)*

  **We agree. However, if we converted the data in this way it would make it more difficult to check the data against the original documents.**

- *Table 2. Could use the units*

  **Good catch! We have updated the table.**

**0.2 Reviewer 2**

- *Please use the full doi designation: https://doi.org/10.5063/f1kk98r5 ? The full designation allowed this reviewer to avoid a search through DataCite to access the KNB Site.*

**The ESSD author instructions say to report a DOI in the abbreviation form (10.5194/xyz). We suspect that this could be hyperlinked to the full designation on typesetting. For example, it appears that the partial doi code provided in the abstract of the pdf manuscript has been hyperlinked to the full doi in the html abstract.**

- *Data very well organized and easily accessible. Very good metadata. Spot checks (Montana, Illinois) showed believable locations and values, evidently quality control has worked reasonably well. Good product, potentially very useful as baseline for both chemical and hydrological / geomorphological purposes.*

- *No information about sampling date in the master .csv file? E.g. a reader gets reference to the report number (.pdf 475, published 1978) and to a page number (for Bloomington Lake, Maclean County, Illinois, actually on page 79 rather than 81 as in .csv file), but no reference to sampling dates. Bloomington Lake data shows nutrient and biological samples collected on 5/11/73, 8/9/73 and 10/17/73. For MacDonald Lake (.pdf 477, page 78 rather than 80 as in .csv) Montana, nutrient and biological sampling on 6/1/75 and 7/28/75. In text we read that sampling of geographic regions occurred by year (e.g. 1973 for southeastern including Illinois and 1975 for western) but the user does not see actual dates where available, or would need to extract those dates themselves? But apparently none of the raw files captured these sampling dates from the original .pdf?*

  **The sampling dates you see are exclusive to the "Biological Characteristics" section, which we did not transcribe, because this has already been done in Stomp et al. (2011). In contrast, the data we report are annual means computed from monthly samples. The NES reports provide no further details about specific sampling dates. We have added clarifying text to the manuscript and metadata on this point.**

- *Data from Illinois resides in two separate sections of .pdf 475 (page numbers 80 to 99 and 100 to 110 contiguous) but one needs to search by storet_code or state name to find all data per each state? This scattering arises from processing sequence?*

  **Yes**

- *Page numbers in .csv file refer to page number of digitized .pdf, not to page numbers used within the individual reports? I did not see reference to this small discrepancy in the metadata.*

  **We have appended additional text to the "pagenum" metadata field. It now reads: "page number of the pdf (not the report page number)".**

- *Need specific clarification about the page number discrepancies and about whether the digitisation process captured the sampling date*

  **See above.**

---

## Author Comment (AC2) · 13 Nov 2017

The responses to each reviewer have been combined in author comment 1 (AC1).

---

## Author Comment (AC3) · 13 Nov 2017

The responses to each reviewer have been combined in author comment 1 (AC1).